# Identification of Old-Growth Mediterranean Forests Using Airborne Laser Scanning and Geostatistical Analysis

Andrea Hevia [†] , Anabel Calzado, Reyes Alejano and Javier Vázquez-Piqué *

Department of Agroforestry Sciences, University of Huelva, Av. Fuerzas Armadas s/n, 21007 Huelva, Spain
* Correspondence: jpique@uhu.es
† Present address: Department Sistemas Físicos, Químicos y Naturales, University Pablo de Olavide, Crta. Utrera km. 1, 41013 Seville, Spain.

**Abstract:** The protection and conservation of old-growth forests (OGFs) are becoming a global concern due to their irreplaceability and high biodiversity. Nonetheless, there has been little research into the identification and characterization of OGFs of the oldest tree species in Mediterranean areas. We used forest inventory data, low-density airborne laser scanning (ALS) metrics, and geostatistical analysis to estimate old-growth indices (OGIs) as indicators of old-growth forest conditions. We selected a pilot area in European black pine (*Pinus nigra* subsp. *salzmannii*) ecosystems where the oldest known living trees in the Iberian Peninsula are found. A total of 756 inventory plots were established to characterize standard live tree and stand attributes. We estimated several structural attributes that discriminate old growth from younger age classes and calculated different types of OGI for each plot. The best OGI was based on mean tree diameter, standard deviation of tree diameter, and stand density of large trees (diameter > 50 cm). This index is useful for assessing old-growthness at different successional stages (young and OGFs) in Mediterranean black pine forests. Our results confirm that the estimation of OGIs based on a combination of forest inventory data, geostatistical analysis, and ALS is useful for identifying OGFs.

**Keywords:** *Pinus nigra*; mediterranean ecosystems; old-growth index; stand structure; geographic information system; forest management

## 1. Introduction

The irreplaceability and high biodiversity of old-growth forests (OGFs) have generated much interest in their identification and maintenance [1], making their protection and conservation a global concern [2,3]. There is, however, no single definition of OGF [4,5]. The author of [6] groups OGF definitions into four categories: (1) definitions that emphasize a lack of disturbance by humans (OGFs have abundant old trees, some of which are approaching the maximum age for the species); (2) definitions that use a minimum age (typically 150 years), combined with the presence of logs, snags, canopy gaps, etc.; (3) definitions that emphasize stand development (in particular, climax forest); and (4) definitions that use an economic threshold (the stands have passed the economic optimum for harvesting). As a consequence, the identification of OGFs could be supported by indicators related to the structural and functional conditions of the forests [7]; focused on the structural attributes and composition, e.g., [8–12], and/or based on the age structure, e.g., [8,13], stand dynamics, and natural regeneration, e.g., [14,15]; or the presence of deadwood, e.g., [16,17].

Key information about sustainable management, conservation strategies, and restoration can be obtained from OGFs, e.g., [3,18,19], since they hold essential data concerning how forest biodiversity has resisted threats [20] over their long lifespan. Unfortunately, a decrease in these unique forests has been observed worldwide [21], as a consequence of deforestation, major disturbances [1], agriculture or active management, and conversion to managed plantations, e.g., [3,22,23]. In Europe, forests are now mainly seminatural, with

undisturbed forests accounting for just 4% of all forest area [24]. Further, only 0.7% of European forests are undisturbed forests composed of native species [21], and just 46% of these forests are strictly protected [25].

An ecological understanding of OGFs requires a multiscale perspective, from individual trees to landscape scales, considering the complexity of forests and their spatial heterogeneity [26] as well as their developmental stages [4]. Moreover, OGFs play an important role in the response to climate change [22], as they continue to sequester carbon for long time periods, but also store more carbon per unit area than any other successional stage [5]. Uncertainty in climate change should be considered when suitably adapting forest management [27], and OGFs may provide valuable information about the resilience of forest ecosystems to climate change [20]. Forest policies and management practices may need to be as diverse as the OGFs themselves [4], taking into consideration the structure and function of the target forests, e.g., [28,29].

In the Mediterranean region, OGF research studies are scarce. It is known, however, that forests in this region have been extensively modified by humans of most countries [20], with OGFs usually found in forest reserves, e.g., [30], or remote mountain areas, e.g., [31], reaching successional stages with high levels of naturalness [32]. Greater efforts should be devoted to identifying and protecting these Mediterranean OGFs, as forests in this region represent the third-richest biodiversity hotspot in the world in terms of plant diversity [33,34]. Despite the relevance of OGFs in Mediterranean areas [31,35,36], most of the studies on OGFs in Europe have focused on forests in temperate and boreal regions, e.g., [37].

In the case of the Spanish *Pinus nigra* Arnold subsp. *salzmannii* (Dunal) Franco forests, with a long history of anthropogenic alteration [38], some remnant old-growth stands with very old individuals can be found in remote mountain areas, e.g., [39], as there has been less silviculture in such inaccessible areas [39]. Studies on the structure of these remaining *P. nigra* OGFs, which contain the oldest known living trees in the Iberia Peninsula, are very limited [27,39,40] and there is an urgent need for more data to guide their conservation and management.

The identification of the best old-growth index (OGI, a dimensionless metric constructed by combining structural features of old forests such as large trees and size diversity) for each forest type and its prediction with spatial modeling could assist in the identification of OGFs in the landscape [11]. Their combination with techniques such as aerial laser scanning (ALS) may be an effective way of generating these OGIs and mapping OGFs [41]. ALS provides information that can be used to predict the three-dimensional structure of vegetation at different scales [42], and this technology has proven useful for estimating a set of attributes related to OGF description; for example, standing dead tree class distributions [43], stand age [44], forest canopy gaps [45], structural canopy complexity [46,47], and forest successional stages [48]. Nonetheless, specific studies describing and mapping old-growth forests using ALS are scarce and none have focused on the Mediterranean region. In other locations, [49] used a random forest framework to model old-growth attributes and predicted an OGI in British Columbia (Canada); [50] investigated the structure of OGFs in the Ukrainian Carpathian Mountains and differences between OGFs and non-OGFs were found over a wide range of ALS metrics; [51] discriminated areas of old growth from areas recovering from selective logging in Sierra Leone (West Africa); and [52] used a combination of airborne LiDAR and satellite imagery to identify and discriminate OGF structures resulting from different disturbance histories in the mixed boreal forest of Quebec (Canada).

The main objectives of this study were: (1) to identify and describe the structural attributes of OGFs; (2) to assess and select the best OGI using field-measured stand variables; and (3) to model the selected OGI using geostatistical analysis and ALS data. We hypothesized that OGIs, which include several attributes and processes associated with OGFs for a given forest type or forest region, might be useful for identifying OGFs and establishing priorities for conservation and management.

## 2. Materials and Methods

### 2.1. The Study Area

This study focused on the southwest portion of the Cazorla Mountains, within the Cazorla, Segura and Las Villas Natural Park, at the northwest of the Andalusian region (Jaén, southeastern Iberian Peninsula; 37°51′ N, 2°52′ W; Figure 1). The climate of the study site is Mediterranean, characterized by severe summer drought and highly variable precipitation between and within years. The average rainfall is 1100 mm year$^{-1}$ (range 400–1900 mm) and the average temperature is 11.7 °C. Pines are the main coniferous trees in the area, with Aleppo pine (*Pinus halepensis* Mill.), Maritime pine (*P. pinaster* Ait.) and black pine (*P. nigra* subsp. *salzmannii*) distributed in slopes and valleys according to edaphic and climatic conditions. Common hardwood species are *Quercus ilex* L. and *Q. faginea* Lam., which grow at lower altitudes and are often mixed with maples (*Acer* spp.), aspens (*Populus* spp.), rowans (*Sorbus* spp.), and ashes (*Fraxinus angustifolia*). Black pine is the most abundant species in the Natural Park, covering 60,000 ha between 1000 and 2000 m [53]. This species can reach 40 m in height and 1.2 m in diameter and is well adapted to poor and shallow soils, steep slopes, and upper and rocky areas, where other more demanding species cannot survive [54]. We selected the *Navahondona* forest (15,588.73 ha) in the Natural Park as a pilot area and focused on the management units of the forest in which *P. nigra* is the dominant tree species (>80%, 4487 ha).

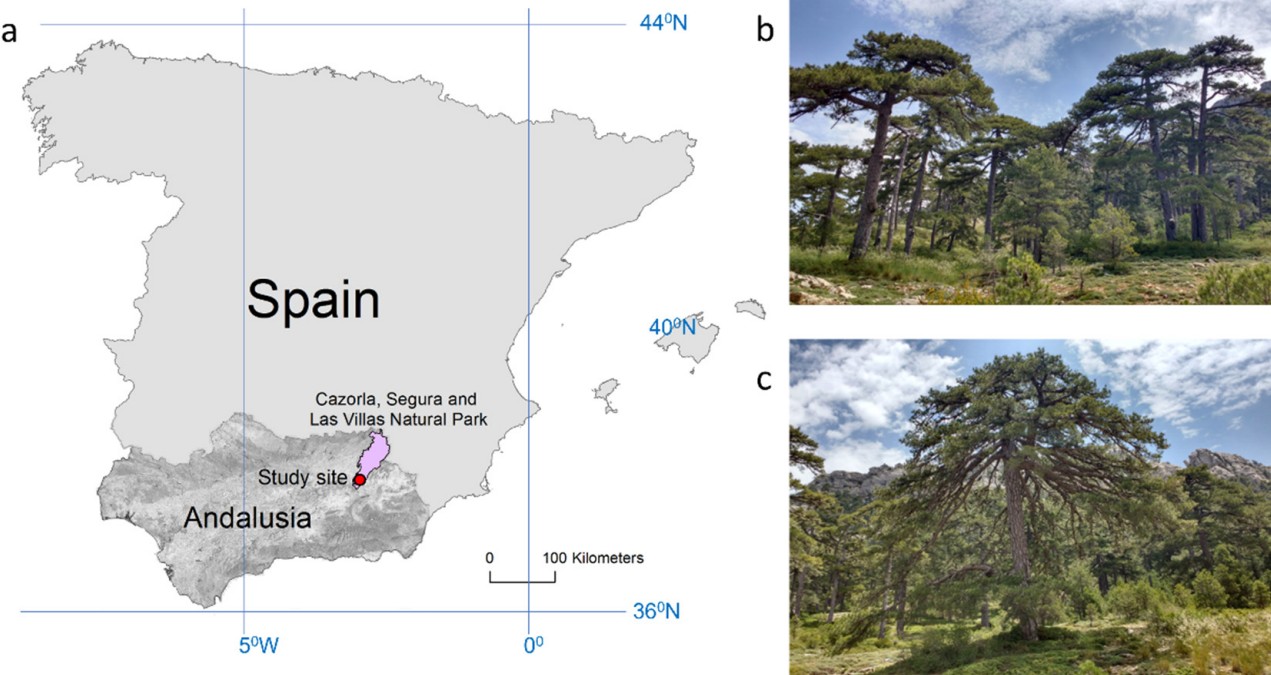

**Figure 1.** (**a**) Study site location (red area) in Cazorla, Segura and Las Villas Natural Park (pink area) in Andalusia, Spain (**a**) left; (**b**,**c**) Physiognomy of old-growth stands in the study area.

### 2.2. Forest Inventory Data

To calculate the OGIs, we used data from management plans on the forest inventory plots (n = 756), consisting of circular plots of 148 m$^2$ (15 m radius) systematically distributed in a grid of 200 m sides resulting in a density of 1 plot per 4 ha. There was no pre-field screening to check if the plots were in forested areas. The inventory included the diameter at breast height (DBH) and total height (HT) of every tree.

### 2.3. Calculation of Old-Growth Indices

An OGI provides a measure of the dissimilarity of a stand relative to young stand conditions [8]. To calculate this index in the inventory plots, we initially followed the

approach of [55] and calculated four structural variables per plot that, according to these authors, successfully discriminate between age classes of a forest: (1) standard deviation of tree DBH; (2) density (trees ha$^{-1}$) of large trees; (3) mean tree DBH; and (4) density (trees ha$^{-1}$) of all trees. These four structural variables can be used to compute an OGI, according to Equation (1) [8]:

$$\text{OGI} = 25 \sum_{i=1}^{4} \left| \frac{x_i - x_{i \text{ young}}}{x_{i \text{ old}} - x_{i \text{ young}}} \right| \tag{1}$$

where *i* represents each of the four structural variables (1–4); $x_i$ is the observed value for the *i*th structural variable; $x_{i \text{ young}}$ is the mean value of the *i*th structural variable for young stands; and $x_{i \text{ old}}$ is the mean value of the ith structural variable for old stands. When the value of any structural variable in a plot is less than that calculated for young stands, the value for young stands applies. Likewise, when the value of the variable exceeds that of the old stands, the value corresponding to an old stand is assigned. Hence, the OGI ranges from 0, when all structural variables correspond to the values of young stands, to 100, when all structural variables correspond to the values of old stands.

The structural values for old stands were determined in a specific inventory carried out in the *Cabañas* forest, situated in the vicinity of *Navahondona* forest. According to our observations and the thoughts of forest managers, the structure, age, and low-intensity management of the stands situated in this forest could be considered representative of old stand structural attributes of *P. nigra* in the area. We established 21 plots of 15 m radius in which *P. nigra* was the dominant species (>80%) and measured the DBH and height of all trees. The structural values for young stands were defined based on the mean values in the 18 inventory plots in which the mean DBH was <20 cm [56,57].

We considered several types of OGI, seeking to account for measures of: (1) the DBH variability in the plot and (2) the definition of large trees. First, as indicators of DBH variability, we calculated two parameters: the standard deviation of tree DBH and the Gini coefficient (GC). GC is a structural heterogeneity index considering the basal area of individual trees in each plot [58], and is computed as follows:

$$\text{GC} = \frac{\sum_{j=1}^{n} (2j - n - 1)\, g_j}{(n-1) \sum_{j=1}^{n} g_j} \tag{2}$$

where *n* is the number of trees in the plot and $g_j$ is the basal area of tree *j* in the plot. Second, for the definition of large trees, we used the thresholds of 50 cm [59], 70 cm [37], and 100 cm DBH [55], and calculated the density of trees (in trees ha$^{-1}$ and basal area (m$^2$ ha$^{-1}$) with DBH > 50, 70, and 100 cm in each plot. As a result, 15 OGIs were calculated, combining different definitions of the aforementioned structural variables (Table 1). Each of the structural variables was compared between old and young forests using analysis of variance.

The most suitable index to be applied in the study area for detecting OGFs was selected by graphical analysis. We plotted the distribution of the indices and compared their boxplots. The pilot area is considered to have significant areas in different successional stages, from young stands to old growth, and this needed to be reflected in an OGI distribution with a large interquartile range covering young and old stages.

*2.4. Geostatistical Modeling*

After selecting the best OGI (Table 1), we analyzed the spatial correlation of OGI data in the pilot area. We used a linear mixed model for testing the significance of spatial correlations considering altitude, slope, and orientation of the inventory plots as model covariates. Covariates were derived from a digital elevation model (DEM) with 5 m horizontal resolution from the Spanish National Centre for Geographic Information (CNIG) (data available at http://centrodedescargas.cnig.es/CentroDescargas/index.jsp#, accessed

on 13 July 2019) and calculated with ArcGis v.10. Orientation was considered a categorical variable with eight categories (N, NE, E, SE, S, SW, W, NW) and altitude and slope continuous variables. The model structure is as follows:

$$OGI_i = \mu + slope_i + altitude_i + orientation_i + e_i \tag{3}$$

where $e_i$ is the error term assuming that $e_i \sim N(0, \sigma^2 + \sigma_1^2)$ and $Cov[e_i, e_j] = \sigma^2[f(d_{ij})]$, with $f(d_{ij})$ a function of the distance between the locations $s_i$ and $s_j$. We chose the spherical distance function $f(d_{ij}) = [1 - 1.5(d_{ij}/\rho) + 0.5(d_{ij}/\rho)^3]$ if $d_{ij} < \rho$, and $f(d_{ij}) = 0$ otherwise. Parameters $\sigma_1^2$, $\sigma^2 + \sigma_1^2$, and $\rho$ correspond to the nugget, sill, and range of the geostatistical model, respectively.

**Table 1.** Structural variables selected to calculate the old-growth index (OGI) in the study area. Structural parameters were weighted to obtain an OGI with 0–100 range. All varieties include the mean DBH of the plot as the structural variable. OGI: old-growth index; STDDBH: standard deviation of tree DBH; GC: Gini coefficient. Density values are in trees ha$^{-1}$ and basal area is in m$^2$ ha$^{-1}$.

| | Structural Parameters | |
|---|---|---|
| **OGI Type** | **DBH Variability** | **Density of Large Trees** |
| 1 | STDDBH | Density of trees > 50 cm DBH |
| 2 | STDDBH | Density of trees > 70 cm DBH |
| 3 | STDDBH | Density of trees > 100 cm DBH |
| 4 | GC | Density of trees > 50 cm DBH |
| 5 | GC | Density of trees > 70 cm DBH |
| 6 | GC | Density of trees > 100 cm DBH |
| 7 | - | Density of trees > 50 cm DBH |
| 8 | - | Density of trees > 70 cm DBH |
| 9 | - | Density of trees > 100 cm DBH |
| 10 | STDDBH | Basal area of trees > 50 cm DBH |
| 11 | STDDBH | Basal area of trees > 70 cm DBH |
| 12 | STDDBH | Basal area of trees > 100 cm DBH |
| 13 | - | Basal area of trees > 50 cm DBH |
| 14 | - | Basal area of trees > 70 cm DBH |
| 15 | - | Basal area of trees > 100 cm DBH |

In order to test the significance of the spatial correlation of OGI data and select the best model variance–covariance structure, we examined the following options before testing the significance of covariates: (1) a model with spatial structure (and considering a nugget effect); (2) a model with spatial structure (no nugget effect); and (3) a model without spatial structure. Models were estimated by the restricted maximum likelihood method and the Akaike information criterion (AIC) was used for model selection [60]. We tested the significance of covariates in the best previous model using all possible combinations of covariates. In this case, models were estimated by maximum likelihood and the best model was selected using the AIC. The statistical analysis was performed with SAS 9.2. In the case of a significant spatial correlation, we predicted the values of the OGI in the study area by kriging with ArcGis v.10 using the predicted values of nugget, sill, and range.

### 2.5. ALS Data Analysis

We used ALS data within the limits of the 756 field plots to compute metrics related to the height distribution and canopy cover. Square 2 × 2 km ALS blocks were obtained from the 2014 CNIG flight data (data available at http://centrodedescargas.cnig. es/CentroDescargas/index.jsp#, last accessed on 28 September 2019). The point cloud was acquired with a maximum of 4 returns per pulse, a theoretical mean density of 0.5 points m$^2$, and vertical root mean square error (RMSE) < 0.20 m. The summary statistics of ALS return density within the plots (pulses m$^{-2}$) were: mean = 0.4, minimum = 0.13, maximum = 2.25, and standard deviation = 0.23.

ALS data were processed using FUSION V 3.50 software [61]. A DEM with a 2 m cell size was generated from classified ground returns and was used to normalize non-ground ALS returns to height above ground surface. The normalized ALS point cloud was clipped, with an independent file generated for each plot (15 m radius). Finally, ALS metrics were extracted for each plot. The ALS metrics employed in this work (Table 2) have been widely used as predictor variables in forest models (e.g., [42,62] low-density ALS and [63] high-density ALS data). Only returns classified as vegetation and with normalized height of between 3 and 32 m were used to compute height and canopy cover metrics (based on field observations).

**Table 2.** ALS metrics computed for each plot. h: tree height (m).

| ALS Metrics | Description |
| --- | --- |
| $h_{mean}$, $h_{mode}$ | mean, mode |
| $h_{min}$, $h_{max}$ | minimum, maximum |
| $h_{SD}$, $h_{CV}$ | standard deviation, coefficient of variation |
| $h_{Skw}$ | skewness |
| $h_{kurt}$ | kurtosis |
| $h_{ID,}$ | interquartile range, |
| $h_{AAD}$ | average absolute deviation |
| $h_{MADmedian}$ | median of the absolute deviations from the overall median |
| $h_{MADmode}$ | median of the absolute deviations from the overall mode |
| $h_{L1}$, $h_{L2}$, $h_{L3}$, $h_{L4}$ | L-moments |
| $h_{Lskw}$ | L-moments of skewness |
| $h_{Lkur}$ | L-moments of kurtosis |
| $h_{01}$, $h_{05}$, $h_{10}$, $h_{20}$, $h_{25}$, ... , $h_{90}$, $h_{95}$, $h_{99}$ | Percentiles |
| CRR | canopy relief ratio: mean height-min height/max height-min height |
| CC | canopy cover: percentage of first returns above 4.5 m/total returns |
| PARA3 | percentage of all returns above 3 m/total all returns |
| ARA3.TFR | ratio between all returns above 3 m and total of first returns |
| PFRAM | percentage of first returns above mean/total all returns |
| PARAM | percentage of all returns above mean/total all returns |
| PARAMO | percentage of all returns above mode/total all returns |
| PFRAMO | percentage of first returns above mode/total all returns |
| ARAM.TFR | ratio between all returns above mean and total of first returns |
| ARAMO.TFR | ratio between all returns above mode and total of first returns |

The best OGI, selected from Table 1, was then used to assess the best ALS metrics for estimating the index. As there is a temporal difference between the ALS data (2014) and the inventory data (2011), we first estimated the DBH of trees in 2014 from the radial increment obtained in the forest inventory. To prevent modeling errors due to a lack of precision in the GPS data in the forest inventory or the impact of silvicultural treatments between the date of the forest inventory and the ALS flight, the 95th percentile of tree height was calculated. Following [64], plots with a difference exceeding 3 m between the two values (inventory and LiDAR) were excluded. As a result, 488 (64.5% of the total) plots were included for estimating OGI values from ALS data.

A multiple linear regression model was used to describe the empirical relationship between OGI and ALS metrics. The general expression is as follows:

$$OGI = \beta_0 + \beta_1 X_1 + \beta_2 X_2 + \ldots + \beta_n X_n + \varepsilon \tag{4}$$

where $X_1$, $X_2$, ... $X_n$ are metrics derived from the ALS dataset (Table 2); $\beta_0$, $\beta_1$, ... $\beta_n$ are the parameters to be estimated; and $\varepsilon$ is an additive error term.

Data were split into two different samples: a random selection of 342 cases (70% of the sample) was used as a training subset, while the 146 unselected cases were used for testing. A stepwise regression method was applied to the training dataset to select independent variables for the model. Although [65] pointed out problems in the use of stepwise selection

in ecology, it has been used successfully in forest modeling from ALS data where the main objective is prediction and not the understanding of a phenomenon, e.g., [42,62,66–69]. The stepwise selection procedure was performed using a combination of forward and backward algorithms implemented in the R Commander package [70] of the R statistical software [71]. We only retained models with no collinearity (VIF < 10) [72] and with all parameters significant ($\alpha = 0.05$). For selecting the best model, we considered the RMSE and adjusted R square ($R^2$adj) statistics. Heteroscedasticity was checked with the Breusch–Pagan test. Finally, RMSE and root mean square error of prediction (RMSEP) were compared to verify that the selected model was not overfitted.

## 3. Results

### 3.1. Selection of Structural Parameters and OGI

All the variables studied showed significant differences between old and young stands (Table 3; $p < 0.05$). Forests in more advanced successional stages (old forests) have a significantly larger mean and standard deviation of DBH, total basal area, and density of large trees above the thresholds of 50, 70, and 100 cm DBH. Moreover, the diameter distribution of forests shows a wider range of distribution in the older stands, with higher tree densities in the 60–80 and 80–100 cm diameter classes (Figure S1). In contrast, total stand density (N, in trees ha$^{-1}$) was significantly larger in the young stands, which also were dominated by smaller diameter trees (<60 cm), unlike the larger trees observed in older stands (Figure S1). Although clear differences have been observed among successional stages, high variability was observed in most of the variables (particularly N and N50 in young forests; see Table 3). Values of basal area and density of trees > 70 and >100 cm were null in early successional stages (young forests).

**Table 3.** Comparison of values (average and standard deviation in parenthesis) of structural variables for old and young stands. Different letters within each row represent significant differences (*p* value < 0.05).

| Structural Variables | Old Forests (*n* = 21) | Young Forests (*n* = 18) | F (*p* > F) |
|---|---|---|---|
| Mean tree diameter (mDBH, cm) | 70.48 (15.0) a | 17.58 (2.01) b | 219.55 (<0.0001) |
| Diameter standard deviation (STDDBH) | 30.89 (17.2) a | 4.28 (2.23) b | 42.45 (<0.0001) |
| Basal area (G, m$^2$ ha$^{-1}$) | 36.08 (15.2) a | 5.74 (5.75) b | 47.88 (<0.0001) |
| Basal area of trees > 50 cm DBH (BA50, m$^2$ ha$^{-1}$) | 34.69 (15.2) a | 1.67 (0.71) b | 91.92 (<0.0001) |
| Basal area of trees > 70 cm DBH (BA70, m$^2$ ha$^{-1}$) | 31.35 (12.0) a | 0 (0) b | 86.98 (<0.0001) |
| Basal area of trees > 100 cm DBH (BA100, m$^2$ ha$^{-1}$) | 27.26 (10.5) a | 0 (0) b | 12.35 (0.0012) |
| Density (N, trees ha$^{-1}$) | 83.15 (41.9) b | 285.30 (277.7) a | 10.88 (0.0022) |
| Density of trees > 50 cm DBH (N50, trees ha$^{-1}$) | 58.94 (29.9) a | 0.78 (3.33) b | 67.00 (<0.0001) |
| Density of trees > 70 cm DBH (N70, trees ha$^{-1}$) | 42.10 (19.6) a | 0 (0) b | 82.30 (<0.0001) |
| Density of trees > 100 cm DBH (N100, trees ha$^{-1}$) | 11.57 (13.3) a | 0 (0) b | 13.59 (0.0007) |
| Gini coefficient (GC) | 0.45 (0.21) a | 0.26 (0.11) b | 11.67 (0.006) |

The total stand density (N) is one of the four structural variables used by [55] for the calculation of their OGI; however, in our study, this variable was not a good indicator of old-growth attributes (Figure 2) since low total density values were found in plots with very low but also very high mean DBH. For this reason, we did not include this structural variable in any of the 15 types of OGI considered (Table 1). The same reasoning can be applied to the total basal area (Figure 3), hence, this variable was also excluded from our OGI calculations.

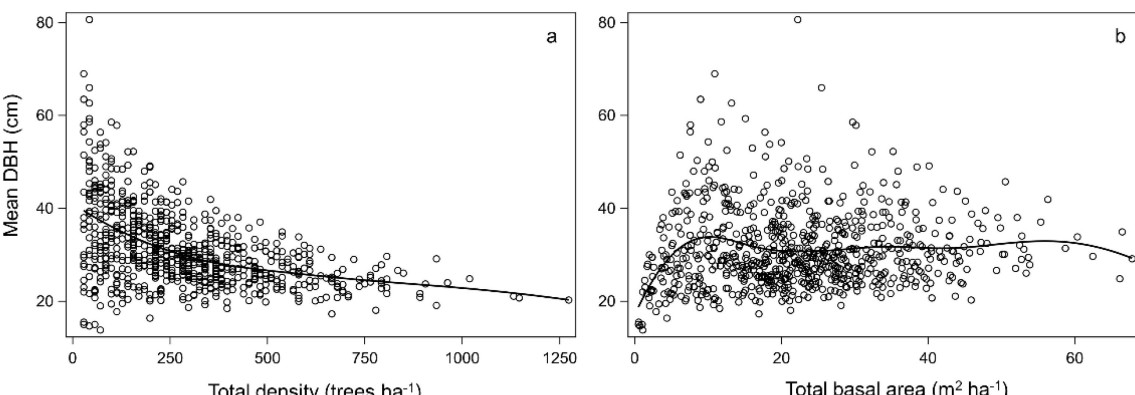

**Figure 2.** Scatterplot of mean diameter at breast height (DBH) and total density (trees ha$^{-1}$) (**a**) and total basal area (m$^2$ ha$^{-1}$) (**b**) at the inventory plots. The curved line shows the tendency (penalized B-spline).

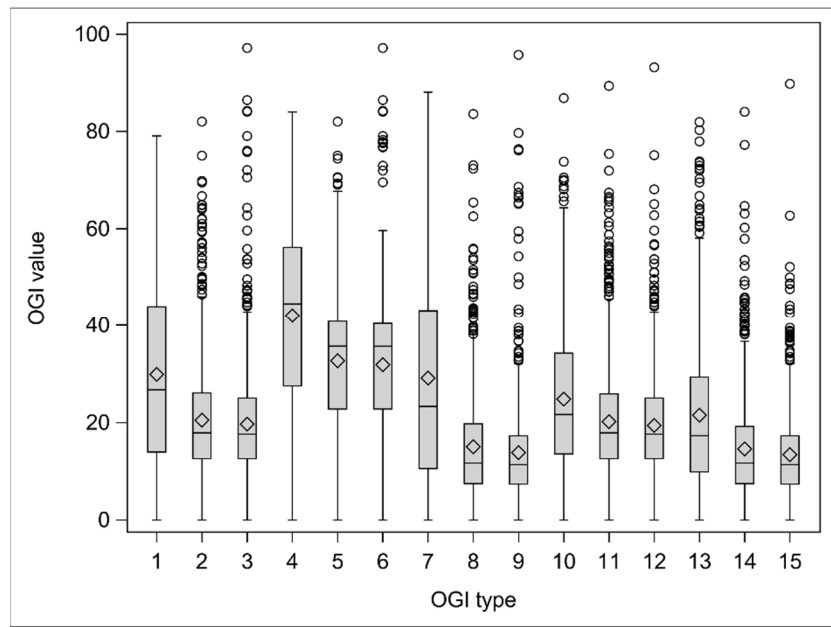

**Figure 3.** Boxplots of the 15 types of old-growth index (OGI) calculated in the study area. The structural variables that are included in each index are described in Table 1.

Boxplots of the 15 OGIs show marked differences in the OGI distribution for the different OGI types (Figures 4 and S2). Clearly, the OGI types with a larger interquartile range are numbers 1, 4, and 7 (hereafter, OGI 1, OGI 4, and OGI 7). These three OGIs are all computed with the mean tree DBH and density of trees > 50 cm DBH (trees ha$^{-1}$), but differ in the structural parameter included to account for DBH variability: standard deviation of tree DBH in OGI 1, GC in OGI 4, and none in OGI 7 (Table 1). The values of OGI 1 and OGI 7 are highly correlated (Pearson r = 0.96) and have a similar frequency distribution (Figure 5), while the distribution of OGI 4 is skewed to higher values and less strongly correlated with OGI 1 and OGI 7 (r = 0.88 and 0.79, respectively). In this case, the GC (included in OGI 4) is not an adequate old-growth attribute because the mean value in the inventory plots (0.418) is very close to the mean value for old stands (0.449; Table 3) and, thus, this parameter will not properly distinguish young and old stands. This is because, in order to calculate the old-growth index, the value of the structural variable (in this case, GC) that is included in the index (see Equation (1)) cannot exceed the characteristic value for old stands (in that case, the characteristic value for old stands applies). In our case, this happens in 38.4% of the inventory plots, which is a very high value. In this high percentage

of plots, the value that enters in the OGI is the same (the characteristic for old-growth stands), so the discriminatory power of using GC as a structural stand characteristic for evaluating old growth is very low.

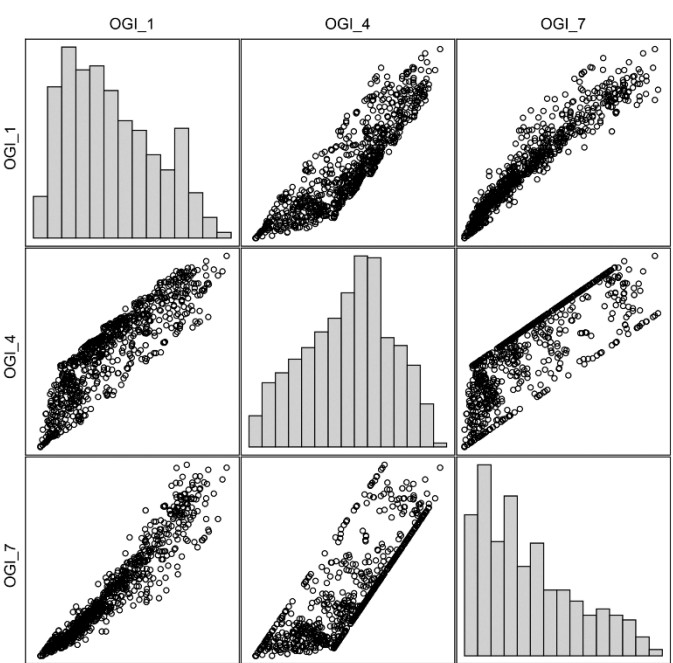

**Figure 4.** Scatterplots (dot plots) and frequency distributions (histograms) of OGIs 1, 4, and 7. The range of the x-axis in the histogram plots is 0–100 and the amplitude of each histogram bar is 7 units. OGI 1 includes the standard deviation of DBH and OGI 4 includes the Gini coefficient, but OGI 7 does not include any tree DBH dispersion parameter. OGI: old-growth index; DBH: diameter at breast height.

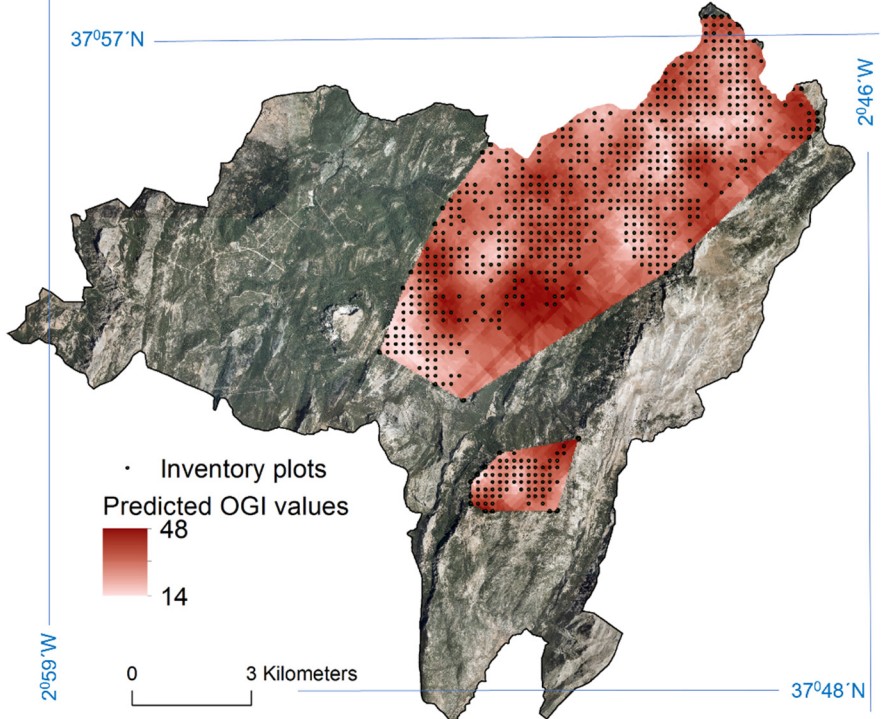

**Figure 5.** Predicted old-growth index (OGI) values in *Navahondona* forest.

The similarity between OGI 1 and OGI 7 reflects the lack of a clear trend in our study area between the size of the trees in the plot (expressed by the mean DBH and density of trees > 50 cm DBH) and the standard deviation of tree DBH (STDDBH). Nevertheless, taking into account the characteristic value of STDDBH, namely, it is much higher in old than young stands (Table 3), we considered STDDBH a desirable structural parameter to include in the OGI of the study forests. Hence, finally, OGI 1 was selected.

### 3.2. Geostatistical Model

The threshold that we used to select our inventory plots (i.e., >80% of *P. nigra* trees) led to the formation of two isolated subareas in our study area (Figure 6) that we called North and South. Given this, in our model selection process, we also considered the hypothesis of having different spatial covariance parameters in these two areas. We finally selected the model with the same spatial covariance structure in both areas (North and South) that incorporates a nugget effect and has no covariates (Table 4, model 5), which is the model with the lowest AIC. The estimated values for nugget, partial sill, and range in the selected model were 315.78, 38.76, and 1579.17, respectively. The slope, orientation, and altitude of the plot were not significantly related to the OGI 1 ($p = 0.22$, $p = 0.51$, and $p = 0.11$, respectively). Therefore, the final selected model (model 5 in Table 4; AIC: 6568.4) was significantly better than the non-spatial model (model 4 in Table 4, AIC: 6585.5), with evidence shown of the spatial correlation in the OGI 1 distribution. Data in Table 4 also support that the most complex model (model 1 in Table 4; AIC: 6575.5, which includes nugget, covariates, and different spatial covariance in areas North and South) does not perform better than the selected simpler model (model 5), which has no covariates and the same covariance in areas North and South.

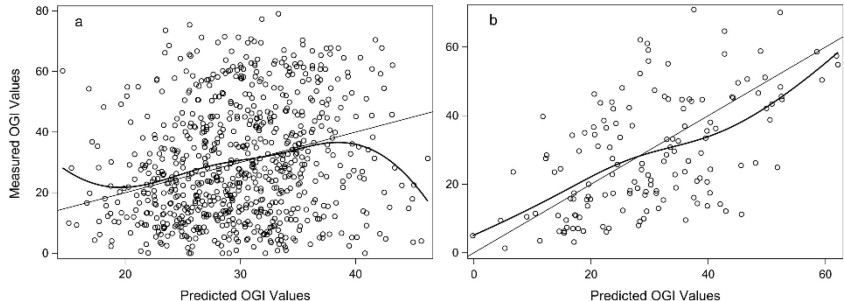

**Figure 6.** Measured old-growth index (OGI) values versus OGI values predicted by the geostatistical model at the inventory plots (**a**) and measured old-growth index (OGI) values versus OGI values from the validation dataset of the ALS model (**b**). The straight line shows a 1:1 relationship, while the curved line shows the tendency (penalized B-spline).

**Table 4.** Geostatistical model selection process. −2LL: −2 Log Likelihood. AIC: Akaike information criterion (the smaller the better). The selected model is model 5.

| Model | Description | −2LL | AIC |
|:---:|:---:|:---:|:---:|
| 1 | With nugget, covariates, and different spatial covariance in areas North and South | 6545.5 | 6575.5 |
| 2 | = Model 1 but same spatial covariance in areas North and South | 6549.4 | 6575.4 |
| 3 | = Model 2 but without nugget | 6751.6 | 6775.6 |
| 4 | With covariates but no spatial covariance | 6563.5 | 6585.5 |
| 5 | = Model 2 but without covariates | 6560.4 | 6568.4 |

The predicted values of OGI 1 in the study area range from 14 to 48 (Figure 6). The OGI 1 is underestimated in plots with high values and overestimated in those with low values.

### 3.3. ALS Model

The model selected for estimating the OGI from ALS data is as follows:

$$\text{OGI} = -30.14945 - 0.25675 \times \text{ARAM.TFR} + 1.90980 \times h_{95} + 8.86526 \times h_{L2} + 32.37033 \times \text{CRR}$$
$$R^2_{adj} = 0.42; \ p \text{ value} < 0.0001; \ \text{AIC} = 2822.4 \tag{5}$$

where ARAM.TFR (a variant of canopy cover) is the ratio of all the returns above the average over the total of all first returns; $h_{95}$ is the 95th percentile; $h_{L2}$ is the second-order moment; and CRR is the canopy relief ratio (describing the degree to which canopy surfaces are in the upper (CRR > 0.5) or lower (CRR < 0.5) portions of the height range) [73]. Figure 6 is a scatterplot of the predicted versus measured OGI 1 values. For this regression model, the RMSE was 14.84 (337 degrees of freedom) and the RMSEP was 14.54. Further information about the model can be seen in Table 5.

**Table 5.** Parameter estimates and goodness-of-fit statistics of the model selected for estimating the OGI 1 from ALS data. ARAM.TFR: ratio between all returns above mean and total of first returns; H95: 95th percentile; $H_{L2}$: second-order moment; CRR: canopy relief ratio.

| Parameter | Estimate | Standard Error | t-Value | $p > |t|$ |
|---|---|---|---|---|
| Intercept | −30.14945 | 5.25588 | −5.736 | <0.0001 |
| ARAM.TFR | −0.25675 | 0.08158 | −3.147 | 0.0018 |
| $h_{95}$ | 1.90980 | 0.38434 | 4.969 | <0.0001 |
| $h_{L2}$ | 8.86526 | 2.12298 | 4.176 | <0.0001 |
| CRR | 32.37033 | 10.86575 | 2.979 | 0.0031 |

This section may be divided by subheadings. It should provide a concise and precise description of the experimental results and their interpretation, as well as the experimental conclusions that can be drawn.

## 4. Discussion

### 4.1. Stand Structure Differences between Young and Old-Growth Forests

Forests of *P. nigra* subsp. *Salzmannii* in the Cazorla, Segura and Las Villas Natural Park reveal a forest structure with a multiple age distribution [39]. The remnant OGFs of this species are characterized by a wide range of tree sizes (Figure S1), with trees attaining notably large dimensions in the case of the largest trees, in agreement with previous studies of this species in the Mediterranean region [39,40]. Only the younger forests are dominated by small-diameter trees, although old sites also have small trees (Figure S1). This could suggest that regeneration has been successful in these sites, providing greater structural heterogeneity to the old-growth stages. Moreover, the older forests have significantly higher total basal area and GC, confirming greater structural heterogeneity. In contrast, younger forests have more simple structures, with a low diversity of sizes and conditions of live trees (Figure S1), as also described in other forest ecosystems [12].

Larger and older trees have been shown to play key ecological roles, not covered by younger and smaller trees [74]. Moreover, older trees provide very valuable ecosystem services (e.g., biodiversity refuge, carbon storage, etc.) [5,75]. Nonetheless, there is a lack of knowledge about *P. nigra* in this respect [27,39], with further research necessary to better understand the role of large trees in biodiversity conservation and sustainable forestry, e.g., [76] in the Mediterranean region. In addition, other attributes of OGFs, such as snags and logs [12], need to be considered.

### 4.2. OGIs to Distinguish Old Growth from Young Forests

Forest structural attributes of OGFs have shown to be useful parameters for identifying forests with OGIs, e.g., [8,77–79]. The OGI selected in this study (OGI 1) integrates three elements of stand structure which reflect the differences in forests across a range of successional stages (young vs. old).

The use of an OGI to evidence structural differences between young and old forests is consistent with previous findings, e.g., [12,78], even when structural attributes only

consider live trees [8]. Nonetheless, such OGIs must be used with caution since they are based on a limited set of measures. In the future, efforts should be made to characterize these forests including as many important OGF structural features as possible [8], such as logs and snags, which play an important role as primary sources of deadwood [80] and sources of energy and nutrients for the ecosystem [18].

Notably, forest structure varies between forest types [12,39,81] and can be modified by disturbances, such as human activities or changes in land use, e.g., [27,39,40]; biotic (insects and pathogens) and abiotic agents (e.g., fire [82]), and wind [15]; and indirect climate effects which also influence the aforementioned forest disturbances [83]. The OGFs studied here are in remote mountain areas where silviculture has been limited, due to difficulty accessing the forests [39], and there is no evidence of disturbance in recent decades. Nonetheless, due to the long lifespan of OGFs, there is a greater probability of a history of disturbance in these forests [84] than in the younger ones. This is even more important in the Mediterranean region where climate change (a driver of disturbances regimes [83]) strongly affect forests [85]. Therefore, more detailed data to reconstruct the past disturbances [79] and climate [31] is required to better understand the present structural attributes of forests and their complexity and dynamics over time.

### 4.3. Geostatistics and ALS to Estimate OGIs

There has been little research into the use of geostatistical analysis and ALS data to estimate OGIs [41], despite the potential usefulness of describing and mapping of OGFs using these data. Models developed are biased for various reasons. Traditional forest inventories require a large number of plots and inventory replications to cover forests in the different successional stages, especially in the most advanced ones (OGFs), and those located in the least accessible areas. On the other hand, we found spatial correlation in OGIs, which could be related to a higher frequency and a more continuous spatial distribution of the earlier successional stages (younger forests). This could indicate that traditional forest inventory is not detecting old-growth characteristics with sufficient detail. Old-growth conditions also vary between forest types and type-specific definitions are necessary [12]. Certain environmental conditions, particularly those that are extreme, may also be associated with distinctive old-growth attributes [12], as occur in the Mediterranean region (characterized by extremely high temperatures and low precipitation). Nevertheless, the presence of spatial correlation in the OGI distribution will be dependent on the forest type and the extension of old-growth forests in the study areas. In fact, even if the geospatial model performed better in our study, this improvement was not very high in comparison with the non-spatial model and this situation can vary in other areas, in which simpler ordinary regression models can perform better. The combination of statistical and geostatistical approaches could also lead to further improvements in OGI prediction [86].

The approach tested in this study using ALS data to define an OGI model can be considered a useful first step but needs refining. Explanatory variables selected were $h_{95}$, $h_{L2}$, CRR, and ARAM.TFR, from which $h_{95}$ and $h_{L2}$ contained the most information about the response. The presence of $h_{95}$ in the model confirms the strong relationship of height with stand age, this being one of the percentiles most useful in modeling stand height using an area-based approach [87]. On the other hand, numerous studies have utilized L-moments for characterizing dasymetric variables with ALS data, e.g., [88–90]. In particular, $h_{L2}$ (i.e., measurement of dispersion similar to standard deviation, with less weight given to outliers [91]) as a predictor variable is in accordance with the results obtained in the characterization of OGFs (such forests showed greater diameter standard deviation). In addition, the OGI selected in this study includes the standard deviation of the diameter.

The model defined to estimate the OGI from ALS data presents a high residual standard error, but it is observed that ALS data and OGI are correlated. These results may be explained by the characteristics of the ALS flight, which was not designed for forest inventory (low density of data of 0.5 first returns m$^{-2}$) [42,62]. Notably, [92] recommended a minimum of 1 pulse m$^{-2}$ (>4 pulses m$^{-2}$ for dense forests on complex terrain) to produce

an operational ALS-based enhanced forest inventory. The results could also be explained by ground plot georeferencing errors, but in the present study, large-size ground plots were used (706 m$^2$); in relation to this, [93] point out that regression analysis using larger plots (>314 m$^2$) appeared more robust to the ill effects of GPS error and therefore, this source of error can be considered less important than the low density of ALS data. The temporal cover provided by PNOA flights has been set at 6 years and new flights will provide a higher density of returns (2 returns m$^{-2}$), which will influence the precision of the relationships between ALS metrics and OGIs; hence, it can be expected that future models estimating OGIs from ALS data will obtain better results. Moreover, future PNOA flights will also allow monitoring of the forests over time.

Further research is needed to obtain more accurate estimations of OGIs using geostatistics and ALS data. A first assessment of the forests, combining these data with those from a forest inventory, could be helpful to recognize the areas in which to conduct a more detailed study to better identify OGFs. Characterization of OGFs using field inventory, ALS data, and OGIs may be useful in future studies, for describing the variation in old-growth attributes and how this variation is spatially distributed, making it possible to generate maps at different scales [41]. Nevertheless, our results must be considered a first approximation, and field inventory data covering forests at different successional stages will be necessary to confirm the OGF distribution and provide essential information to preserve and manage these valuable ecosystems.

## 5. Conclusions

The OGI that combines the structural variables of mean DBH, standard deviation of tree DBH, and density of trees with diameter > 50 cm proved to be a good index for assessing old-growth attributes in Mediterranean *Pinus nigra* forests, making it a valuable tool for their identification and characterization and easy to calculate from standard forest inventories. The Gini coefficient as a measure of tree DBH diversity was less useful than the standard deviation in this species. The geostatistical OGF model proved the existence of spatial correlation in the OGFs' attributes and we succeeded in providing a map of the distribution of the OGFs, in which the location of old-growth stands was not related to altitude, exposition, or slope. In this sense, spatial prediction of OGIs derived from the geostatistical model provides a good first step in screening old-growth characteristics on forested lands. The ALS model highlighted that the main variables related to old-growth attributes were the 95th percentile of height and the second-order moment. The ALS model's performance was limited by the low density of the ALS data, but it is expected that this circumstance will be overcome shortly with the incorporation of data with higher resolution. We consider finally that the methodology and workflow presented in this study could be applied for detecting OGFs of other conifer and broad-leaved species in different ecosystems, to monitor changes in OGFs' attributes over time, and to establish priorities for conservation and management.

**Supplementary Materials:** The following supporting information can be downloaded at: https://www.mdpi.com/article/10.3390/rs14164040/s1, Figure S1: Diameter size distributions of live trees from old (black bars) and young (grey bars) stands; Figure S2: Old-growth indices (OGIs) for each OGI type calculated in the study area. Normal (blue line) and kernel density (red dashed line) of the distribution of each index. The structural variables included in each OGI type are described in Table 1.

**Author Contributions:** Conceptualization, A.H., R.A., A.C. and J.V.-P.; methodology, A.H., R.A., A.C. and J.V.-P.; formal analysis, A.H., R.A., A.C. and J.V.-P.; investigation, A.H., A.C., R.A. and J.V.-P.; resources, R.A. and J.V.-P.; writing—original draft preparation, A.H., A.C., R.A. and J.V.-P.; writing—review and editing, A.H., A.C., R.A. and J.V.-P.; visualization, A.H., R.A., A.C. and J.V.-P.; supervision, A.H., A.C., R.A. and J.V.-P.; project administration, R.A. and J.V.-P.; funding acquisition, R.A. and J.V.-P. All authors have read and agreed to the published version of the manuscript.

**Funding:** This work was supported by the following projects: "Iberian Heritage Project", funded by the Netherlands Organization for Scientific Research (NWO, project number 236-61-001), National Geographic Society-Waitts Grant Program ("Millennia old black pines and Andalusian Cultural Heritage to unravel human-environment interactions in the Western Mediterranean", W329-14), the Biodiversity Foundation of the Ministry of Agriculture and Fisheries, Food and Environment ("Bosques viejos frente al cambio climático. Vulnerabilidad, capacidad adaptativa e implicaciones frente a la gestión forestal", PRCV00433) and Ministry of Economy, Industry and Competitiveness (MINECO) ("El final del ciclo envejecimiento, mortalidad y regeneración en pinares mediterráneos, y su papel en la adaptación ante un ambiente en cambio (OLDPINE), AGL2017-83828-C2-2-R). The Ministry of Agriculture and Environment of the Regional Government of Andalusia provided the AF forest inventory data. AH have been supported by PinCaR project (UHU-1266324, FEDER Funds, Andalusia Regional Government, Consejería de Economía, Conocimiento, Empresas y Universidad 2014-2020).

**Data Availability Statement:** The dataset generated during and/or analyzed in the current study is available from the corresponding author on reasonable request.

**Acknowledgments:** We are grateful to Teresa Moro from the Natural Park, and Valentin Badillo from the Cazorla, Segura and Las Villas Natural Park, for their interest and support. The forestry engineering students Raúl García-Raga and Carlos Maeztu (University of Huelva), and Alex Boninsegna (University of Padova) contributed to the fieldwork as part of their final thesis undergraduate studies.

**Conflicts of Interest:** The authors declare no conflict of interest.

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
