# Peer review of "Identification of Old-Growth Mediterranean Forests Using Airborne Laser Scanning and Geostatistical Analysis"

_remotesensing, doi:10.3390/rs14164040_

Round 1

Reviewer 1 Report

Andrea Hevia, Anabel Calzado, Reyes Alejano and Javier Vázquez-Piqué present a readable, diligently planned, executed and interpreted work, based on an airborn laser scanning of pinus stands in a Mediterranean protected area (Spain). It provides new insights on the possibility to estimate and discriminate whether or not a Pinus nigra forest can be classified and recognised as an old growth one. The work is absolutely valuable and merits publication on Remote Sensing. In fact, just very minor suggestions can be focused:

after line 138, formula (1): the upper summation bound should be just 4, not i=4, as it is in the following formula (2)

lines 157- 159: why the Gini coefficient has been chosen? Reference [8] in table 1 presents a number of diversity quantifications; is Gini index the best performing one in this case?

after line 159, formula (2): this formula, just like the cited one [58], although correct, could be better formatted: the (n-1) term in the denominator should be carried out of the summation, unlike the (2j-n-1) which depends on the summation index j.

Although the authors honestly and clearly indicates that their analytical procedure deserves further testing and application, some interesting points could be risen on the possibility to specifically apply to old growth forests dominated by broad leaved species. For instance, in the Mediterranean region there are several interesting cases of Fagus sylvatica OGFs which would provide other relevant remote sensing experimental examples.

Author Response

Dear reviewer,

Thank you very much for your constructive revision and positive comments on the manuscript. We provide in the following lines a point-by-point response to the comments:

Reviewer comment 1: "after line 138, formula (1): the upper summation bound should be just 4, not i=4, as it is in the following formula (2)".

Response 1: we have changed the formula.

Reviewer comment 2: "lines 157- 159: why the Gini coefficient has been chosen? Reference [8] in table 1 presents a number of diversity quantifications; is Gini index the best performing one in this case?".

Response 2: We have tested two different indexes regarding the DBH variability: the standard deviation of tree DBH and the Gini coefficient (lines 157-158, Table 1). The finally selected Old-Growth index did not include the Gini coefficient, as it is explained in lines 291-293 of the manuscript (the mean value of the Gini coefficient in the inventory plots (0.418) is very close to the mean value for old stands (0.449, Table 3) and, thus, this parameter will not distinguish properly young and old stands).

Reviewer comment 3: "after line 159, formula (2): this formula, just like the cited one [58], although correct, could be better formatted: the (n-1) term in the denominator should be carried out of the summation, unlike the (2j-n-1) which depends on the summation index j."

Response 3: we have changed the formula.

Reviewer comment 4: "Although the authors honestly and clearly indicates that their analytical procedure deserves further testing and application, some interesting points could be risen on the possibility to specifically apply to old growth forests dominated by broad leaved species. For instance, in the Mediterranean region there are several interesting cases of Fagus sylvatica OGFs which would provide other relevant remote sensing experimental examples."

Response 4: Thank you for your comment. We have included the following comment at the end of the conclusions section: "We consider finally that the methodology and workflow presented in this study could be applied for detecting OGFs´of other conifer and broad leaved species in different ecosystems, to monitor changes in OGFs attributes over time and to establish priorities for conservation and management."

Reviewer 2 Report

This article is very interesting and very well structured, showing important findings on the topic under analysis, specifically in methodological terms, being the results obtained relevant for the scientific community.

Besides the strong theoretical framework on which this work is cast, the results are well supported by the data analysis.

The text is clear and, properly drafted. I have no rectifications to suggest.

The figures and tables seem to be appropriate.

The fundamental element of this paper is the methodology, which was extensively exposed and explained.

In the discussion, the last part of “Implications for management” is too much abstract. The author should present some more specific proposals and not just refer some generalistic ideas based in “common sense”.

The conclusions are very vague, not focused in the confirmation of the objectives initially proposed. Authors should improve the conclusions, synthetizing their main findings.

Nevertheless, I consider the contents of this paper to be of good quality. It is very clear in terms of the methodology employed, which seems to be appropriate, and reveals a clear and logical structure.

Author Response

Dear reviewer,

Thank you very much for your constructive comments and the general positive opinion on the manuscript. We include now a point-by-point response to the comments:

Comment 1: In the discussion, the last part of “Implications for management” is too much abstract. The author should present some more specific proposals and not just refer some generalistic ideas based in “common sense”.

Response 1: We agree with the reviewer that this section of the discussion is very abstract and that, in fact, it is not supporting any of the results or methodological sections of the manuscript. The importance of the old-growth forests has been already highlighted in the introduction to justify this study. In this conditions we think that it is better to delete this section of the manuscript completely and the corresponding references 96, 97, 98 and 99.

Comment 2: The conclusions are very vague, not focused in the confirmation of the objectives initially proposed. Authors should improve the conclusions, synthetizing their main findings.

Response 2: We have rewritten completely the conclusions section to focus better in the main findings and objectives.

Reviewer 3 Report

Authors have done great work in characterizing old-growth forests using OGI's and the methods they adopted are also not complex and easy to implement by users or management agencies. Although I also agree that more structural parameters will be required in the future to improve these models. But I enjoyed reading the manuscript and suggest minor spell checks and editing throughout the paper. 

The manuscript discusses the identification of old-growth forest focusing on the Mediterranean region. The authors have generated OGIs using different combinations of matrices and claimed that the OGIs can be useful in assessing old growth forests at different successional stages especially for the black pine forest. 

The topic is original and authors have performed novel research in addressing research gaps in the field of identification and tracing of old growth forest. As in the recent times, there has been a significant decline in the OGFs, the approach used in this study might help conservation experts to track the changes and composition of OGFs in different parts of the world. The manuscript is overall a good read although I have few suggestions which I have listed below:

  1. Please try to improve the resolution of the figures and maintain a standard dpi. Figure 1(a), please consider changing this figure as it is clearly visible that the satellite image is pasted over the  shape file of Spain. Also consider adding a lat long grid in fig 1(a) and fig 5.
  2. In the discussion section, authors should also compare and discuss present results with the papers from the past. 
  3. The conclusion section needs a revision, authors should focus on concluding their results and the significance of their research. Currently most of the literature in this section focuses on methodology.

Author Response

Dear reviewer,

Thank you very much for your constructive revision and the overall positive comments on the manuscript. We provide now a point-by-point response to the suggestions:

Reviewer comment 1: Please try to improve the resolution of the figures and maintain a standard dpi. Figure 1(a), please consider changing this figure as it is clearly visible that the satellite image is pasted over the  shape file of Spain. Also consider adding a lat long grid in fig 1(a) and fig 5.

Response 1: We have modified Figure 1(a) and we have included a long lat grid in figures 1a and 5. Regarding the other figures of the manuscript, the quality is good when the original file is opened.

Reviewer comment 2: In the discussion section, authors should also compare and discuss present results with the papers from the past.

Response 2: As far as we know, this is the first work focused in the elaboration of an old-growth index in mediterranean mountain ecosystems and in the modelling of such an index by geostatistics or Aisbone Laser Scanning data. In the introduction section (lines 86-95) we include information on other studies that have used similar approaches but the ecosystems were so different (e.g. mixed boreal forest in Quebec or British Columbia, forrsts in Sierra Leona or Ukraine) that results and findings were not comparable at all.

Reviewer comment 3: The conclusion section needs a revision, authors should focus on concluding their results and the significance of their research. Currently most of the literature in this section focuses on methodology.

Comment 3: We have rewritten completely the conclusions section to better focus on our results.